# Awareness and Use of Canine Quality of Life Assessment Tools in UK Veterinary Practice

**DOI:** 10.3390/ani13061105

**Published:** 2023-03-20

**Authors:** Claire Roberts, Emily J. Blackwell, Emma Roe, Joanna C. Murrell, Siobhan Mullan

**Affiliations:** 1Bristol Veterinary School, University of Bristol, Langford BS40 5DU, UK; emily.blackwell@bristol.ac.uk (E.J.B.); siobhan.mullan@ucd.ie (S.M.); 2School of Geography and Environmental Sciences, University of Southampton, Highfield Campus, Univesity Road, Southampton SO17 IBJ, UK; e.j.roe@soton.ac.uk; 3Highcroft Veterinary Group, 615 Wells Road, Bristol BS14 9BE, UK; jo.murrell@cvsvets.com

**Keywords:** dog, quality of life, veterinarian, welfare, assessment

## Abstract

**Simple Summary:**

Assessing the quality of life (QOL) in dogs is difficult but formal assessment tools exist, often in the form of owner-completed questionnaires. Use of these tools in veterinary practice has been recommended by various veterinary associations. This study investigated current awareness and use of canine QOL assessment tools in veterinary practice in the UK. An online survey was completed by 90 veterinary surgeons and 20 veterinary nurses. One third were aware of the existence of canine QOL assessment tools, but less than four percent were using one in practice. Most vets and nurses were willing to use one as a tool, but reported that lack of time and potential resistance from owners were barriers to use. Results suggest that QOL assessment tools developed by researchers are not well distributed to veterinary professionals, and that several barriers inhibit their use.

**Abstract:**

The use of formal canine quality of life (QOL) assessment tools in veterinary practice has been recommended. An online survey investigated awareness, use and barriers to use of these tools in the UK. An anonymous 24-question survey was advertised through veterinary groups and social media. Ninety veterinary surgeons and twenty veterinary nurses responded. Thirty-two respondents (29.1%) were aware of the existence of formal canine QOL assessment tools. Of the three tools listed, current use was less than four per cent. No statistically significant influence of respondent age, role (veterinary surgeon or nurse) or possession of additional qualifications was found on the awareness of QOL tools (*p* > 0.05). Over half of respondents (55.5%) would ‘certainly’ or ‘probably’ be willing to use a QOL assessment tool. The main barrier to use was lack of time. Other barriers included a perceived resistance from owners. Although current use and awareness of canine QOL assessment tools in UK veterinary practice is low, veterinary professionals appear willing to use the tools within their daily practice. This discrepancy implies that QOL assessment tools are not well disseminated to veterinary surgeons and nurses in practice and that various barriers inhibit their use.

## 1. Introduction

Quality of life (QOL) is an important topic in the welfare of companion animals, encompassing many aspects of their life [1]. It has been defined as ‘an individual’s satisfaction with its physical and psychological health, its physical and social environment and its ability to interact with that environment’ [2]. In humans, the assessment of one’s own QOL is subjective and is based on the wants, needs, feelings and opinions of the individual [3,4]. For companion animals, the inability to self-report leads to the necessity of a human proxy, for example the owner or a veterinary surgeon. This inherently makes assessment more difficult and less accurate. Quality of life is also multidimensional, including physical, emotional and social aspects of life [3]. This dictates a global assessment of the life of the companion animal rather than just its physical health, which may not fall within the traditional expertise of veterinary professionals trained in animal health. 

Formal assessment of QOL is becoming increasingly discussed in companion animal medicine; within recent years both the British Veterinary Association and the World Small Animal Veterinary association have highlighted the importance of welfare assessment in veterinary practice, including the application of practice-based quality of life assessment tools [5,6]. There are many potential uses for a standardised measure of quality of life in veterinary practice which include, but are not limited to, decision making during euthanasia, improving care and evaluation of treatment, and evaluating welfare in the home [7].

Several companion animal QOL assessment tools already exist in the veterinary literature, and to this point have mainly focused on dogs. A systematic review up to July 2013 revealed 11 peer-reviewed canine QOL assessment tools with initial validity testing and a further 41 tools without [2]. In addition, mainstream tools have been developed and disseminated, such as the PDSA’s Pet-Wise MOT [8] and the PetDialog app [9]. Although Belshaw, Asher, et al. [6] described minimal subsequent use of some validity-tested tools by researchers, to our knowledge no study has described the use of QOL assessment tools in the wider UK veterinary population. Several studies have explored the awareness and use of QOL assessment tools amongst medical professionals, mainly in oncology [10,11,12] and paediatrics [13,14]. These studies featured a survey-based methodology.

The aim of this study was to use a survey to investigate the awareness of veterinary surgeons and nurses in the UK of the existence of formal canine quality of life assessment tools (validated or otherwise), the current levels of use of these tools within veterinary practice, and the willingness to use these tools in the future. Additionally, barriers to the use of QOL tools in veterinary practice were evaluated.

## 2. Materials and Methods

A previous study investigating paediatric clinicians’ perspectives on quality of life [13] was found to have almost identical aims to the present study. The paediatric focus, which involves the use of proxies, relates well to veterinary medicine. The survey had also been based on several previous studies. The survey was adapted for relevance to veterinary practice and dogs, and for readability where necessary. Demographic questions were added from the Royal College of Veterinary Surgeon’s (RCVS) surveys of veterinary surgeons [15] and nurses [16] to compare the respondents with the general UK veterinary populations. The survey was piloted, resulting in no changes (Appendix A).

The survey was available online between February and May 2018. Eligible respondents were self-selected veterinary surgeons and nurses who carried out consultations with dogs in a veterinary practice in the UK. The RCVS register was not available for research purposes at the time, so links to the survey were shared in veterinary literature, on social media and through veterinary groups. An incentive of entry into a prize draw for a GBP 50 voucher was given. The survey was titled ‘Dog Consultation Survey’ to avoid bias towards those with an interest in quality of life.

The survey consisted of 24 questions, plus three additional questions where veterinary surgeons and veterinary nurses were separated. Questions included familiarity with the existence of canine QOL assessment tools in general plus the awareness and current/past use of three specific tools alongside various pain assessment tools. The three QOL tools were listed to include a pain-focused tool (Canine Brief Pain Inventory) [17], a relatively recent global tool competed on paper (PetWise MOT) [18] and an online tool (PetDialog app) [9]. Pain scores included an acute pain score (Glasgow Composite Pain Scale) [19] and three chronic pain scores (Canine Orthopaedic Index, Helsinki Chronic Pain Index, Liverpool Osteoarthritis in Dogs scale) [20,21,22,23]. Willingness to use a QOL assessment tool in some aspect of practice was assessed on a likert-type scale. Fourteen potential barriers to the use of QOL tools in practice were listed with the option of ‘not a barrier’ ‘somewhat of a barrier’, ‘moderate barrier’ and ‘strong barrier’.

Chi-squared analyses were used to investigate any association of age (16–25, 26–35, 36–45, 46≤), role (veterinary surgeon or nurse) or possession of additional qualifications with familiarity and potential use of QOL assessment tools. Chi-squared analyses were also conducted to assess potential associations between the awareness of formal QOL assessment tools with a belief they could be used in a veterinary clinic and willingness to use in a veterinary clinic. Bonferonni corrections set the p-value at 0.016 for both sets of analysis.

IBM SPSS Statistics Version 24.0 was used for analyses [24]. The study was granted ethical approval by the University Health Sciences Faculty Research Ethics Committee.

## 3. Results

### 3.1. Demographics

There were 111 responses to the survey. One participant did not answer any demographic questions and was excluded from analysis where demographics were required. No other exclusions were made. Twenty were veterinary nurses (18.2%) and 90 (81.8%) were veterinary surgeons. Fifteen were male (13.6%) and 95 female (86.4%). A comparison of the sample population with the wider veterinary population from the 2014 RCVS surveys of vets [15] and nurses [16] can be seen in Table 1. The present study had a larger proportion of female vets and a lower proportion of nurses with additional qualifications than the RCVS survey. The average age of veterinary surgeons and nurses in the present study was lower than in the RCVS population.

### 3.2. Awareness and Use of QOL Assessment Tools

Of the 110 respondents who answered the question, 32 (29.1%) were aware of the existence of QOL assessment tools for dogs, with a further fifteen (13.6%) ‘not sure’. Veterinary professionals’ awareness and use of the specific QOL tools listed in the survey can be seen in Table 2. Awareness of pain assessment tools, including the Glasgow Composite Pain Score and the Liverpool Osteoarthritis in Dogs (LOAD) tool was higher than that of QOL assessment tools.

Fewer than four percent of respondents reported they were currently using one of the three stated QOL assessment tools, with none currently using the PetDialog app. In comparison, 10% and 56% of respondents were currently using the Liverpool Osteoarthritis in Dogs scale (LOAD) assessment tool and the Glasgow Composite Pain Score, respectively.

### 3.3. Willingness to Use QOL Assessment Tools

Most of the respondents (93/111; 83.8%) reported it was completely their own choice whether to use a QOL assessment tool. Most also expressed confidence that they had the skills and knowledge necessary to use a QOL assessment tool (92/109; 84.4%). If it were assumed that a valid and reliable tool was available, over half reported that they would surely (20; 18.2%), or probably (41; 37.3%) want to use it in some aspect of their practice. A further 35.5% selected maybe (*n* = 39). Nine percent reported they would probably not (*n* = 10), and no respondents selected ‘certainly not’. However, fewer respondents reported that they actually planned to use an assessment tool, with less than a third selecting surely (11; 10.1%), or probably (21; 19.3%). A further 48.6% (*n* = 53) selected maybe, with nineteen respondents selecting ‘probably not’ (17.4%) and five ‘certainly not’(4.6%).

### 3.4. Subgroup Analysis

No statistically significant associations were detected between respondent age (16–25, 26–35, 36–45, 46≥), role (veterinary surgeon or nurse) or possession of additional qualifications with the awareness of QOL tools (all *p* > 0.05).

There was no association between being familiar with QOL tools and a belief that they could be used in a veterinary clinic setting. There was a significant association found between familiarity with tools and planning to use X^2^ (1) = 12.1 (*p* < 0.001) tools in the future (surely/probably will versus maybe/probably not/certainly not). Familiarity with tools was not significantly associated with wanting to use tools using the corrected *p*-value X^2^ (1) = 5.3 (*p* = 0.02).

### 3.5. Barriers to Use of QOL Assessment Tools

Figure 1 shows to what extent aspects of using QOL assessment tools were reported as potential barriers in clinical practice. Time for completion of the tool itself was most frequently reported as a ‘strong barrier’, followed by time to discuss the tool and time to enter results into clinical notes. Data on consultation length showed most respondents had a 10 (39; 35.1%) or 15 (59; 53.2%) minute consultation length, with one respondent having flexible 10–15 minute appointments, four percent having 20 minutes (*n* = 3.6), five professionals (4.5%) from referral centres with over half an hour per consultation and two (1.8%) from out-of-hours centres who do not have set consultation times. A perceived resistance from owners in completing a tool was also a ‘strong barrier’. Insufficient knowledge about QOL and ‘your own priorities are different’ were the aspects most reported as ‘not a barrier’.

## 4. Discussion

To our knowledge, this is the first study to investigate veterinary professionals’ awareness, use and potential barriers to use of canine quality of life assessment tools in UK veterinary practice. Despite several published quality of life assessment tools [6], and the recent addition of mainstream tools like the PetWise MOT, fewer than a third (29.1%) of veterinary professionals who responded to the survey were aware of their existence. The number of veterinary surgeons and nurses reporting currently using the QOL assessment tools listed was even lower,, although they did report a willingness to use them.

These results suggest that QOL assessment tools are not being well disseminated to veterinary surgeons and nurses working in practice. No associations were found between age, role and possession of additional qualifications with awareness of QOL assessment tools, indicating that this is a profession-wide issue. QOL tools are more widely used in human medicine [13,25]; QOL assessment tools have certainly existed longer in human than veterinary medicine, but it could also be that QOL tools are better disseminated in medical practice. Literature and conferences are reported as the main places doctors obtain information on QOL assessment tools [14]. With 6500 veterinary professionals attending BSAVA congress in 2019 [26], veterinary conferences could be an ideal location for QOL tool dissemination. Publication in mainstream veterinary literature rather than difficult to access journals may also be beneficial. Knowledge of the existence of tools seems to be associated with planning to use them, so increasing awareness is the first step in encouraging use.

Very low use of QOL assessment tools for dogs was reported, at less than four percent for the three tools in the survey. The global scales (PetWise MOT and PetDialog) were less well known than the pain-related CBPI, which is discussed below. Over half of respondents reported they would be ‘certainly’ or ‘probably’ be interested in using a QOL assessment tool in their practice, with none reporting they would not. However, less than a third reported that they planned to do so. This discrepancy between willingness and planning to use is reflected in the medical profession [10,14]. Most respondents felt they had both the necessary skills and the power to introduce QOL assessment if they wanted to. It is likely that other barriers are preventing use.

As with the medical profession [10,14,27], time was the main barrier reported for using QOL assessment tools. In a recent qualitative study, veterinary surgeons reported an issue of time restrictions in preventative healthcare consultations, especially with very young or old animals [28]. Vets also discussed that ten-minute consults are not long enough to discuss more than immediate problems [28], as would be necessary for a formal QOL assessment. In the current study, most respondents had 15-minute appointments, with only 4% having 20 minutes. Although referral centres had longer consultation times, they are also likely to need more in-depth discussion of a specific health issue and may also have time pressures. Although a recent study where a canine QOL assessment tool was completed by clients in the waiting room showed that it did not impact consultation time [29], this does not address the additional time barriers of administration and training. Additionally, for owners to complete the forms in the waiting room without any veterinary input may introduce the potential for bias from caregiver burden bias, which has been shown to impact QOL assessment [30,31]. QOL scores completed by owners may also be impacted by their age [32].

After consultation time constraints, perceived resistance from owners was the strongest barrier. Again, this reflects human medicine, where doctors expect poor patient compliance [10] and some patients have demonstrated inability to complete a QOL questionnaire [24]. Vets have discussed feeling that most owners would not be interested in ‘extras’ in their consultations and would only want to talk about the specific reason for the visit [28]. However, in the same study owners reported appreciating more thorough and lengthy consultations [28]. One owner even stated they wished there were a “questionnaire with… some ticky boxes”, which matches the structure of some QOL assessment tools. Dog owners have also identified the value of a QOL assessment during a veterinary consultation [28]. Assumptions about owners by vets may therefore not match with the actual feelings of the owners. Additionally, there is no information about the awareness of canine QOL issues in the owner population.

Most of the veterinary professionals in the current survey reported insufficient information on tools and availability of tools as ‘somewhat of a barrier’ and a ‘moderate barrier’, respectively. The lack of knowledge of tools reported above might imply that availability of tools would be a strong barrier, as it is for doctors [13,14]. This finding may be related to the use of the likert-type scale ranging from ‘not a barrier’ to ‘strong barrier’; if availability of time is such a big issue, other barriers might seem small in comparison. Qualitative data such as interviews or a case study might serve to investigate barriers more deeply.

The comparison between awareness and use of QOL scoring tools with that of pain scoring tools is interesting, as evaluation of chronic pain and assessment of QOL share similar features [33]. The LOAD tool and the Glasgow Composite Pain Score were both well known and widely used. As mentioned above, the CBPI, a chronic pain-centric QOL tool, was the most well known of the three QOL tools listed. Pain scales have existed longer than QOL scales; the Glasgow Composite Pain Score [19] and CBPI [17] were first published in in 2007. Again, qualitative data collection could explore this further by investigating how veterinary professionals have become aware of these tools and why they use them.

This study has some limitations. The sample size is small, so all statistical analyses should be treated with caution; there is a potential that associations may exist but have not been detected. Comparison of the study population with the RCVS surveys suggests results may not be generalisable to the wider veterinary professional population, although the RCVS surveys include vets and nurses who do not work with dogs such as farm/equine practitioners and retired professionals. The study sample contained a majority of female-identifying respondents, reflecting the RCVS’s reported nurse population but less similar to that of veterinary surgeons. A significant gender difference on concern for animal welfare has been demonstrated, with female vets finding certain aspects of the five freedoms, such as environment, more important than male vets [34] and female veterinary students displaying more concern towards animal welfare than males [35]. There is a potential for this gender bias to have skewed towards a greater interest in quality of life.

Distribution of the survey was restricted because of unavailability of the RCVS register and insufficient funds to use a commercial mailing company. With these methods, a more substantial response rate may have been possible.

## 5. Conclusions

Formal assessment of canine quality of life in veterinary practice using standardised tools could have numerous and wide-reaching benefits. Despite the existence and continued production of canine QOL assessment tools, this study highlights a lack of awareness of these tools in veterinary practice and low levels of their use. Veterinary professionals appeared willing to use QOL assessment tools for dogs in their practice, but face several barriers including lack of time and a perceived resistance from owners. Awareness of such barriers could assist future and past tool developers in better dissemination amongst the profession.

## Figures and Tables

**Figure 1 animals-13-01105-f001:**
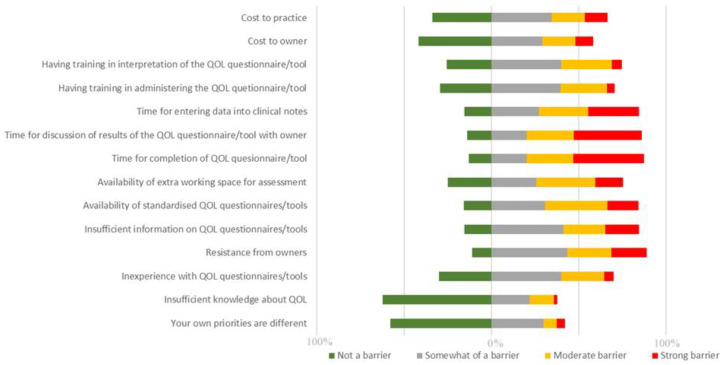
Diverging bar chart showing the percentage of veterinary professionals rating each barrier to the use of quality of life (QOL) assessment tools on a likert-type scale.

**Table 1 animals-13-01105-t001:** Demographics of the veterinary professionals who completed the online questionnaire for the present study, and the corresponding demographics from the 2014 RCVS surveys of Veterinary Surgeons and Nurses.

	Current Survey			2014 RCVS Survey	
	Veterinary Surgeons *n* (%)	Veterinary Nurses *n* (%)	Total *n (%)*	Veterinary Surgeons (%)	Veterinary Nurses (%)
Gender Male Female	14 (16) 76 (84)	1 (5) 19 (95)	15 (14) 95 (86)	(46.2) (53.8)	(2) (98)
Years of work 0–5 6–10 11–15 16+	35 (39) 22 (25) 8 (9) 24 (27)	8 (40) 5 (25) 2 (10) 5 (25)	43 (39) 27 (25) 10 (9) 29 (27)	N/A	N/A
Additional qualifications Yes No/no response	38 (42) 42 (47)	11 (55) 9 (45)	49 (45) 51 (46)	(44.4) ^1^ (55.6) ^1^	(26) ^1^ (74) ^1^
Age 16–25 26–35 36–45 46–65 ≥66	11 (12) 43 (48) 17 (19) 14 (16) 4 (4)	8 (40) 6 (30) 4 (20) 2 (10) 0 (0)	19 (17) 49 (45) 21 (19) 16 (15) 4 (4)	Average age 44.3	Average age 30.6
Type of practice Small first opinion Mixed Referral Out of Hours	67 (74.4) 16 (17.8) 5 (5.6) 2 (2.2)	13 (65.0) 5 (27.8) 2 (10.0) 0 (0)	80 (72.7) 21 (19.1) 7 (6.4) 2 (1.8)	(69.2) ^2^ (20.4) ^2^ (10.5) ^2^ N/A	(70.5) ^2^ (18.0) ^2^ (11.5) ^2^ N/A

^1^ Veterinary-related qualifications only. ^2^ Adjusted % to include only those who work in a practice with dogs. N/A = data not available.

**Table 2 animals-13-01105-t002:** Number and percentage of veterinary professionals surveyed who are aware of and who use/have used selected pain and quality of life (QOL) scoring/assessment tools.

	Unaware of *n* (%)	Aware of but Not Used *n* (%)	Use Currently *n* (%)	Have Used *n* (%)
Canine Brief Pain inventory ^1^	76 (69.1)	26 (23.6)	4 (3.6)	4 (3.6)
Canine Orthopedic Index ^2^	73 (67.6)	31 (28.7)	2 (1.9)	2 (1.9)
Glasgow Composite Pain Scale ^2^	3 (2.7)	23 (20.7)	62 (55.9)	23 (20.7)
Helsinki Chronic Pain Index ^2^	81 (73.6)	21 (19.1)	5 (4.5)	3 (2.7)
Liverpool Osteoarthritis in Dogs Scale ^2^	66 (60.6)	28 (25.7)	11 (10.1)	4 (3.7)
PDSA PetWise MOT ^3^	90 (81.8)	16 (14.5)	1 (0.9)	3 (2.7)
Zoetis PetDialog ^3^	90 (82.6)	16 (14.7)	0 (0)	3 (2.8)

^1^ Pain-related QOL assessment tool. ^2^ Pain assessment tool. ^3^ QOL assessment tool.

## Data Availability

The data presented in this study are available on request from the corresponding author. The data are not publicly available due to ethical restrictions.

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
