# Peer review of "Awareness and Use of Canine Quality of Life Assessment Tools in UK Veterinary Practice"

_animals, 2023, doi:10.3390/ani13061105_

Round 1

Reviewer 1 Report

In general this is an important field: Animal welfare considerations which should be more highly ranked in the interest of professionals than they are currently.

Thus my remark of average interest to the readers is more a criticism of public opinion (as expressed in the results of the survey) than of the authors.

Some minor remarks:

in the introduction and discussion, some brief reference to contemporary concepts of animal welfare science (needs vs necessities, does the animal have what it wants AND is it healthy, 5 freedoms) could be made

The tools/questionnaires they are comparing should be included in the supplemnents section in order for the readers to evaluate them

It looks to me as if some results/numbers are used repeatedly in several statistical tests ( e.g. when calculating significancies both for ALL participants and for subgroups). In these cases a Bonferroni-correction or similar procedure would be necessary

.In the discussion, when suggesting that dog holders might fill in the forms, the problem of self-reporting bias (a common theme in social etc sciences) might briefly be addressed.

Author Response

Many thanks to Reviewer one for their time and helpful comments to help improve this manuscript. We have addressed the following points:

Point one: in the introduction and discussion, some brief reference to contemporary concepts of animal welfare science (needs vs necessities, does the animal have what it wants AND is it healthy, 5 freedoms) could be made

Response: we intend to publish information on a study regarding the definition of quality of life and its relationship to the concepts of animal welfare, so will comment on this in our future manuscript and hope this is acceptable

Point two: The tools/questionnaires they are comparing should be included in the supplements section in order for the readers to evaluate them

Response: we have now properly listed and referenced the tools used in our survey in the main body of the text to allow easier access to them if required and to properly credit their developers

Point three: It looks to me as if some results/numbers are used repeatedly in several statistical tests ( e.g. when calculating significancies both for ALL participants and for subgroups). In these cases a Bonferroni-correction or similar procedure would be necessary

Response: We have used a Bonferroni correction and have changed the significant results and their discussion as necessary

Point four: In the discussion, when suggesting that dog holders might fill in the forms, the problem of self-reporting bias (a common theme in social etc sciences) might briefly be addressed.

Response: A good addition, we have added commented on the potential for bias from caregiver burden when owners comment on their dog’s quality of life.

Reviewer 2 Report

Comments on the manuscript “Awareness and use of canine quality of life assessment tools in UK veterinary practice” submitted to the Animals

General Comments

I appreciate the opportunity to review the manuscript “Awareness and use of canine quality of life assessment tools in UK veterinary practice”.

In the manuscript, there is a description and analysis of knowledge about QOL and the willingness of veterinarians and nurses to apply them. From the responses of a sample of 110 respondents, there is little knowledge about the questionnaires, and a small portion effectively applies the QOL. It is concluded that the lack of time of the vets and nurses and the lack of adherence of the owners are the main barriers to the more widespread use of the QOL.

The study is exciting and original and may be informative for anyone studying the veterinary sciences. The writing has a well-structured logic and is easily readable.

The method is well described, in such a way that other researchers will be able to replicate the study, even in languages other than English.

The statistical approach is simple but acceptable. Large and complex statistical operations are not necessarily a sign of a good study.

The list of references is adequate to expose the problem and support the arguments exposed in the text.

Despite these positive aspects, there is a flaw in the structure of the text. In the end, there is no clear conclusion in the manuscript. The authors should emphasize which one or which conclusions of the study.

Below, I point out some doubts and minor errors that the authors should consider:

Line 40:  I believe that the QOL response is also the result of the observations of respondents who are in the veterinary profession. The answers are not necessarily just "feelings and opinions (random)".

Table 1: I believe the correct sign is ≥66.

What is OOH? Please, explain.

Table 2: I couldn't understand why all the variables in the lines add up to 100%, but two variables (the Canine Brief Pain Inventory and the Glasgow Composite Pain Score) sum is less than 100%. I think there is an error, or the authors did not explain how they concluded at these percentage values in the variables the “Canine Brief Pain Inventory” and the “Glasgow Composite Pain Score”.

Line 149: necessary, not “necessary”.

Line 160: The correct notation is  46≥ (It is not 46≤).

Line 262: Some studies suggest differences between men and women in the perception of well-being, affections, and relationships between pets and humans (e.g., Herzog, 2015; Mariti et al., 2017). In a sample with 84% to 95% women (vet surgeons and nurses, respectively), there may be a gender bias. This bias is a limitation of the study, so it should be included in the discussion.

References

Herzog, H. A. (2007). Gender differences in human–animal interactions: A review. Anthrozoös, 20(1), 7-21.

Mariti, C., Giussani, S., Bergamini, S. M., & Gazzano, A. (2017). Attitude towards pets in veterinary surgeons: a comparison between female and male veterinarians in Italy. Dog behavior, 3(2), 17-20.

Author Response

Many thanks to Reviewer teo for their time and helpful comments to help improve this manuscript. We have addressed the following points:

Despite these positive aspects, there is a flaw in the structure of the text. In the end, there is no clear conclusion in the manuscript. The authors should emphasize which one or which conclusions of the study.

Response: Apologies for this omission when formatting the manuscript, the conclusion has been added

Line 40:  I believe that the QOL response is also the result of the observations of respondents who are in the veterinary profession. The answers are not necessarily just "feelings and opinions (random)".

Response: we have clarified that this relates to self-assessment of QOL in humans

Table 1: I believe the correct sign is ≥66.

Response: thank you, this has been corrected

What is OOH? Please, explain.

Response: we have expanded this acronym

Table 2: I couldn't understand why all the variables in the lines add up to 100%, but two variables (the Canine Brief Pain Inventory and the Glasgow Composite Pain Score) sum is less than 100%. I think there is an error, or the authors did not explain how they concluded at these percentage values in the variables the “Canine Brief Pain Inventory” and the “Glasgow Composite Pain Score”.

Response: thank you for noticing a typo in the table, this has been corrected. The Glasgow Composite Pain Scale Sum appears to be correct in our version

Line 149: necessary, not “necessary”.

Response: thank you, this is now corrected

Line 160: The correct notation is  46≥ (It is not 46≤).

Response: thank you, this is now corrected

Line 262: Some studies suggest differences between men and women in the perception of well-being, affections, and relationships between pets and humans (e.g., Herzog, 2015; Mariti et al., 2017). In a sample with 84% to 95% women (vet surgeons and nurses, respectively), there may be a gender bias. This bias is a limitation of the study, so it should be included in the discussion.

Response: a very interesting and pertinent factor, we have commented on this and will be aware of this gender bias for future manuscripts. Many thanks for the relevant references